# Air Quality and Residents' Health in China: An Empirical Analysis Based on Spatial Simultaneous Equations

Hua He and Zining Li *

School of Science, Hebei University of Technology, Tianjin 300401, China
* Correspondence: lizining@126.com

**Abstract:** Large-scale air pollution has an impact on the health and travel of residents in China. First, this article uses the gravity model and finds that China's air quality presents a typical spatial correlation network. The network structure has changed from complex to simple, indicating that China's air quality is gradually improving. Second, this paper uses spatial simultaneous equations to study the effect of air quality on residents' health. The results show that an increase in the air quality index of local and surrounding provinces will have a significant impact on the health of residents. The absolute and relative air quality affects the lives of residents. The decline in the health of residents from the previous period will prompt the government to adopt various pollution reduction policies, which will improve air quality in the current period. Based on the empirical research conclusions, this article makes policy recommendations.

**Keywords:** air quality; spatial correlation network; residents' health; spatial simultaneous equations

## 1. Introduction

Industrial waste gas, automobile exhaust gas, and the residential burning of coal cause serious air pollution. In January 2013, news networks and China began to report that many places in China suffered from severe haze, which affected residents' travel and health and caused a series of impacts on people's lives [1–3]. Words such as $PM_{2.5}$ and $PM_{10}$ are gradually becoming familiar to the Chinese people. The incidence rate of respiratory diseases, cardiovascular diseases and a series of other diseases has increased, and the life and health of residents are seriously threatened [4–6].

In 2016, General Secretary Xi Jinping proposed at the National Health and Health Conference: "Without the health of the whole people, there would be no all-round well-off", "Health is the inevitable requirement of promoting people's all-round development", and "A good ecological environment is the foundation of human survival and health", emphasizing the importance of national health. In 2017, the two sessions proposed the blue-sky defense war. In 2018, the State Council had formulated *the three-year action plan for winning the blue-sky defense war.* The Chinese government formulated a series of policies to improve air quality. Under the compulsory management of the central to local governments at all levels, China's air pollution has been effectively alleviated [7,8].

However, the "2020 Global Environmental Performance Index (EPI) Report" shows that among 180 countries, China's air quality ranks 137th [9]. Protecting the ecological environment is still one of the problems that China needs to solve. How much does air quality affect the health of residents? As residents pay more attention to health, how much impact does residents' health have on air quality? These two issues have become the focus of this article. The paper uses spatial simultaneous equations to study the impact of air quality on the health of residents. The empirical results can provide data support for government departments to formulate policies.

## 2. Literature Review

The relevant literature on the effects of air pollution on residents' health mainly focuses on the following two aspects.

The first aspect is the exposure-response relationship research in the field of epidemiology and environmental science. Epidemiological studies have found that air pollution has a negative effect on chronic lung disease [10] and causes cardiovascular and cerebrovascular diseases [11]. Air pollution is closely related to morbidity and mortality in the population [12–14]. The increase in major air pollutants will increase the relative risk of a number of respiratory diseases [15–17].

Environmental scientific research has found that the toxicity of persistent toxic pollutants in the atmosphere has toxic effects on early embryonic development, cardiovascular system development, and nervous system development [18]. Some scholars have found that polycyclic aromatic hydrocarbon (PAH) pollution in fine atmospheric particles has certain potential health risks to community residents [19–21]. The carcinogenic risk of heavy metals in $PM_{2.5}$ inhaled by the population varies with the seasons, and the overall pattern is winter > spring > autumn > summer [22]. The carcinogenic risks are mainly As, Cr, Cd, and Co, and the carcinogenic risk to adults is greater than that to children [23–25].

The second aspect is the study of influencing factors in economics. The health impact of air pollution is a health production function pioneered by Grossman [26]. Cropper [27] and Alberini [28] added air pollution variables to the theoretical model of the health production function and believed that the decay rate of health capital depends on the level of air pollution, thus establishing an analytical framework for the impact of air pollution on health. Most Chinese scholars have found that air pollution has a negative impact on the public health of residents [29–31]. Kim [32] found that $PM_{10}$ not only caused an increase in the number of daily respiratory hospital visits but also increased additional healthcare costs. Air pollution has a significant impact on residents' health [33,34], and there is regional heterogeneity in the eastern and central regions [35,36]. The direct impact of air pollution on health expenditure varies among different population groups. The health consumption expenditure of rural residents is less than urban residents; the health consumption expenditure of male residents is less than female residents; and the health consumption expenditure of less educated residents is less than that of highly educated residents [37].

In summary, different scholars have studied the impact of air pollution on residents' health from different perspectives. However, most studies have overlooked a problem: the degree of personal pollution exposure is endogenously determined [38], and the single-equation estimation method will lead to biased estimates of the relationship between air pollution and health. This paper solves this problem by using spatial simultaneous equations. This paper makes two contributions. The first is that it uses the gravity model to study the spatial correlation network of China's air quality, to explore the development law of China's air quality, and to clarify the strategic choice of air pollution control. The second is that it uses spatial simultaneous equations to not only solve the problem of the endogeneity of explanatory variables but also to solve the problem of the spatial correlation of variables. Through the study of spatial simultaneous equations, this paper found that air pollution has different degrees of impact on residents' health. With residents paying more attention to their health, they will, in turn, increase their sensitivity to air pollution, thus providing experience and reference.

## 3. Spatial Correlation Network Analysis of Air Quality in China

### 3.1. Research Methods of Spatial Correlation Network

In recent years, Chinese scholars have mostly used the gravity model to analyze the spatial correlation network [39–41]. The gravity model originated from universal gravitation in physics and is now widely used in economics, management, humanities and

social sciences. The gravity model not only considers geographical factors but can also analyze the dynamic evolution trend of air quality. Its calculation formula is,

$$Grav_{ij} = k_{ij}\frac{A_i A_j}{D_{ij}^{\ b}}$$
$$k_{ij} = A_i / (A_i + A_j)$$

(1)

where, $Grav_{ij}$, $k_{ij}$ and $D_{ij}$ respectively represent the gravity force of the air quality between province $i$ and province $j$, the contribution coefficient of air quality relationship, and the shortest distance between governments. $A_i$ and $A_j$ represent the air quality index of province $i$ and province $j$. $b$ is the distance friction coefficient. When $b$ equals 1, the formula is more realistic and can more reasonably reflect the intensity of the air quality correlation among provinces. When the gravity value from province $i$ to province $j$ is greater than the average gravity value, the air quality of province $i$ will affect the air quality of province $j$. When the gravity value from province $i$ to province $j$ is less than the average gravity value, the air quality of province $i$ will not affect the air quality of province $j$.

### 3.2. Spatial Correlation Network Characteristics of Air Quality in China

The Air Quality Index (AQI) is objective, and most scholars use this indicator to represent air quality [42–44]. Like most scholars, this paper uses the Air Quality Index to measure China's air quality. The network nodes are 30 provinces in China (excluding Xizang, Hong Kong, Macao and Taiwan), and the time span of the samples is 2014–2019. The air quality index comes from the China Meteorological website.

According to the value calculated by Formula (1), this paper uses Gephi0.9.2 to draw the spatial correlation network of air quality. The results are shown in Figures 1–4. This paper only shows the spatial correlation network of air quality in 2014, 2016, 2018, and 2019. The larger the network node is, the greater the connectivity. That is, the air quality of a province with this characteristic has more influence on the air quality of other provinces. The average continuous output degree in 2014 was 9.069, the average continuous output degree in 2016 was 6.375, the average continuous output degree in 2018 was 4.917, and the average continuous output degree in 2019 was 4.174. From Figures 1–4, it can be found that the spatial correlation of air quality presents a typical network structure. Figure 1 shows more provinces and relationships, and the network structure represented in this figure is the most complex. With the continuous control of air pollution, the number of provinces and relationships in Figure 4 is greatly reduced, and the network structure is gradually simplified in China.

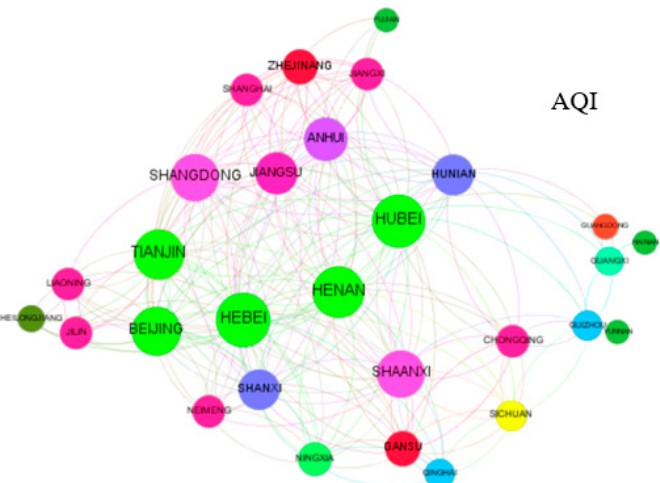

**Figure 1.** AQI spatial correlation network in 2014.

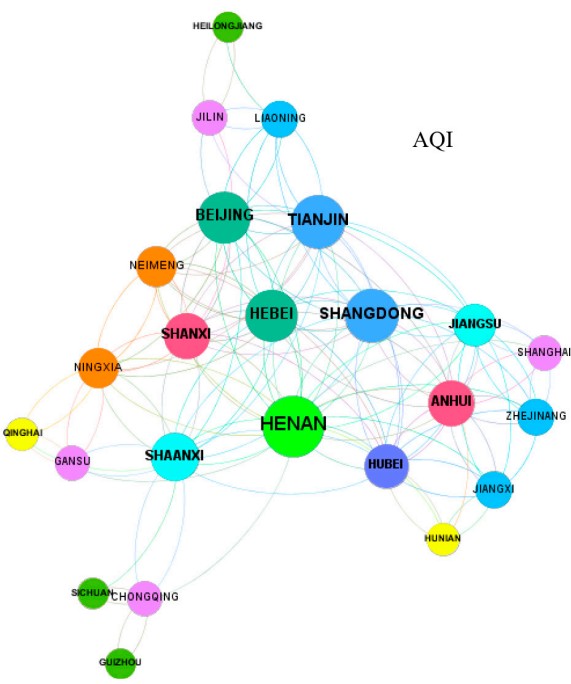

**Figure 2.** AQI spatial correlation network in 2016.

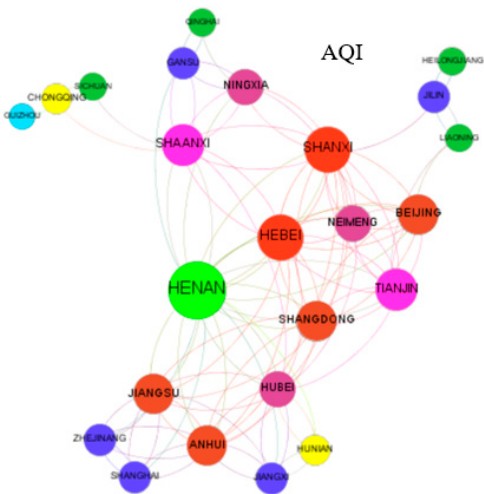

**Figure 3.** AQI spatial correlation network in 2018.

Theoretically, there are 29 relationships between the air quality of a province and the air quality of other provinces, and the maximum number of relationships is 870 (30 × 29). From Figure 1, the total number of spatial relationships of air quality in China in 2014 was 263. From the size of the nodes, the provinces with the most serious air pollution started from Beijing, Tianjin, Hebei, Shanxi and Shandong and had gone all the way south, involving provinces such as Hubei, Anhui, Jiangsu and Shanghai. Among them, the air quality of Hebei was the worst. According to the measurement of the gravity model, its air pollution had spread to 22 provinces. In addition to Hebei Province, air pollution in Henan and Hubei had also spread to 20 or more provinces. The air quality of Beijing, Tianjin, Shanxi, Jiangsu, Anhui, Shandong, Hunan and Shaanxi had affected more than 10 provinces. From Figure 2, the total number of spatial relationships of air quality in China in 2016 was 153. Compared with 2014, the overall air quality index began to decline in 2016, and the air pollution index had dropped by an average of 16.73%. The impact of a province on air pollution in surrounding provinces had reached 20 provinces, but this was no longer the case. The air quality of Beijing, Tianjin, Hebei, Jiangsu, Zhejiang, and

Anhui had affected more than 10 provinces, but the number of affected provinces had also dropped considerably. For example, the air quality in Beijing affected 18 provinces in 2014, but in 2016 it affected only 12 provinces. The most obvious change was with Hebei, where the air quality in 2014 affected 22 provinces, but in 2016 had dropped to 12 provinces.

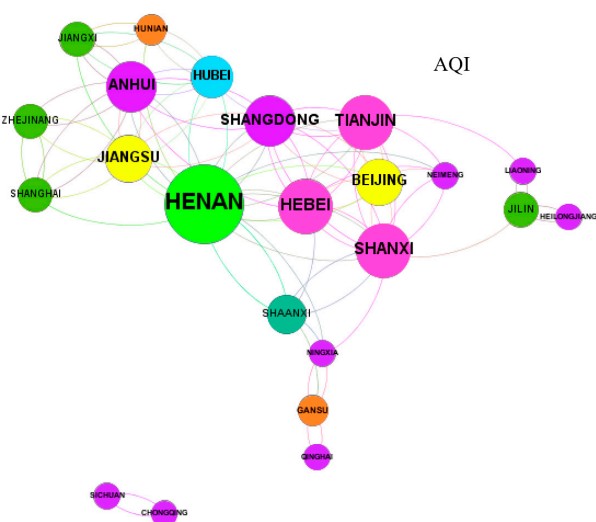

**Figure 4.** AQI spatial correlation network in 2019.

From Figures 3 and 4, compared with 2014, China's air pollution had further decreased in 2018 and 2019. In 2018, China's air quality index had dropped by 22.71%, and in 2019, China's air quality index had dropped by 28.05%. In 2018, the total number of the largest spatial correlations of air quality in China was 118. Air quality in Henan had affected 16 provinces. The air quality of Hebei and Shanxi had affected 10 provinces, and the air quality of other provinces had affected fewer than 10 provinces. In 2019, the total number of the largest spatial correlations of air quality was 96, and the number of air quality correlations had been further reduced. China's air quality had improved. Air quality in Henan had affected 14 provinces, and the air quality of the other provinces had affected fewer than 10 provinces. The air quality index of Hebei, the center of serious air pollution, had decreased by 72.73% in 2019 compared with 2014.

## 4. Analysis of the Impact of Air Quality on Residents' Health in China

### 4.1. Variable Description

The endogenous variables are air quality (AQI) and resident health (Reh). The air quality index describes the cleanliness of the air, which consists of six major pollutants. The larger the value is, the worse the air cleanliness. Residents' health is an important symbol for evaluating the prosperity of a nation and a country. Most scholars use self-assessed health to measure residents' health [45–47]. Some scholars, such as Xie [48] and Peng et al. [49], have used prevalence and population mortality to measure residents' health, respectively. There are three main indicators used internationally to measure the health of residents, namely, life expectancy, maternal mortality, and infant and child mortality. According to the literature and international standards, combined with the actual situation in China, this paper uses the average number of medical visits of residents to measure the health of residents (time). In robust testing, the fatality rate is used to represent the health of residents.

The control variables of the residents' health equation are the proportion of the urban population (Pup), the volume of domestic garbage collected and transported (Dgv), and ecological restoration and governance (Erg). The larger the proportion of the urban population is, the higher the urbanization rate, and the higher the average income level of residents [50]. This can lead to an improved level of living medical care which is beneficial to the improvement of residents' health. However, this can also lead to an increased

volume of domestic garbage which would occupy a large amount of land, pollute soil, air and water bodies, and destroy the ecological environment; the untimely removal and transportation of domestic garbage would have an impact on resident health and lead to disease [51,52]. Ecological restoration and management are conducive to environmental protection and high-quality economic development [53]. Annually, China invests many financial resources in ecological restoration, hoping to continuously explore and innovate in ecological restoration and create a suitable ecological environment for its population.

The control variables of the air quality equation are energy consumption intensity (Eci), per capita GDP (Pgdp), the proportion of secondary industry (Psi), and technology investment (Tei). Energy consumption intensity is calculated by dividing energy consumption (10,000 tons of standard coal) by regional GDP (100 million yuan). An increase in energy consumption intensity reduces the contribution to green economic efficiency [54]. The higher the per capita GDP is, the higher the economic level. The more attention that is paid to environmental quality, the more conducive it is to ecological environmental protection [55] and to the health of residents. The industrial structure can effectively improve energy efficiency and reduce environmental pollution [56], and the optimization of the proportion of secondary industry can promote green economic development and effectively improve air quality [57,58]. If science and technology investment is used in clean and environmental protection technology innovation, air quality will be improved [59–61].

### 4.2. Research Method

To analyze the interaction between endogenous variables, referring to the literature of Kelejian [62], this paper constructed a cross-regional spatial simultaneous equation model.

$$\mathbf{Y_n} = \mathbf{Y_n}\mathbf{B} + \mathbf{X_n}\mathbf{C} + \overline{\mathbf{Y}}_\mathbf{n}\mathbf{\Lambda} + \mathbf{U_n} \tag{2}$$

where, $\mathbf{Y_n} = (y_{1n}, y_{2n}, \cdots, y_{mn})$, $\mathbf{X_n} = (x_{j1n}, x_{j2n}, \cdots, x_{jkn})$, $\mathbf{U_n} = (u_{1n}, u_{2n}, \cdots u_{mn})$, $\overline{\mathbf{Y}}_\mathbf{n} = (\overline{y}_{1n}, \overline{y}_{2n}, \cdots, \overline{y}_{mn})$, and $\overline{y}_{jn} = \mathbf{W}_n y_{jn}, j = 1, \cdots, m$. $y_{jn}$ represents the observed value of the endogenous variable in region n in the $j^{th}$ equation. $x_{jln}(l = 1, 2, \cdots k)$ denotes the observed value of the $l^{th}$ exogenous variable in the region n in the $j^{th}$ equation. $u_{jn}$ represents the random disturbance term in region n of the $j^{th}$ equation. $\mathbf{W}_n$ represents the weight matrix. $\mathbf{B}, \mathbf{C}$ and $\mathbf{\Lambda}$ represent the parameter matrix of $m \times m$, $k \times m$ and $m \times m$.

This paper analyzed the impact of air quality on residents' health, using panel data.

According to the variable description and Formula (1), the spatial panel simultaneous equation is,

$$Peh_{it} = \beta_1 w_{ij}AQI_{jt} + \beta_2 w_{ij}Peh_{jt} + \beta_3 AQI_{it} + \beta_4 Pup_{it} + \beta_5 Dgv_{it} + \beta_6 Erg_{it} + c_i + \gamma_t + u_{it} \tag{3}$$

$$AQI_{it} = \alpha_1 w_{ij}AQI_{it} + \alpha_2 Peh_{it-1} + \alpha_3 Eci_{it} + \alpha_4 Pgdp_{it} + \alpha_5 Psi + \alpha_6 Tei + \delta_i + \lambda_t + v_{it}^0 \tag{4}$$

where, $i, j$ represent different provinces, $i, j = 1, 2, \cdots, n$, and $i \neq j$. $t$ represents time, $t = 1, 2, \cdots, T$. $\beta_1, \beta_2, \beta_3, \beta_4, \beta_5, \beta_6$ and $\alpha_1, \alpha_2, \alpha_3, \alpha_4, \alpha_5, \alpha_6, \alpha_7$ indicate the parameters to be estimated. $w_{ij}$ indicates the spatial weight. When two provinces are not adjacent, $w_{ij} = 0$. When two provinces are adjacent, $w_{ij} = 1$. $c_i, \delta_i$ represent individual fixed effects. $\gamma_t, \lambda_t$ indicate time fixed effects. $u_{it}, v_{it}$ represent a random disturbance term.

Estimating Formulas (3) and (4) will produce a simultaneous equation deviation. Therefore, this paper adopted a systematic estimation method, which is called 3SLS.

### 4.3. Empirical Analysis of Air Quality on Resident Health

#### 4.3.1. Data Sources

The research sample was made up of 30 provinces in China (excluding Xizang, Hong Kong, Macao and Taiwan) as the research object, the time span was 2014–2019, and the number of samples was 180. Data regarding the total energy consumption were obtained from the *China Energy Statistical Yearbook.* Data regarding the average number of medical

visits of residents, fatality rate, proportion of urban population, volume of domestic garbage collected and transported, ecological restoration and governance, per capita GDP, the proportion of secondary industry, and technology investment were obtained from the *China Statistical Yearbook.* The software used for data analysis was Stata 17.0.

### 4.3.2. Descriptive Analysis of the Impact of Air Quality on Residents' Health

Figure 5 shows a scatter plot of air quality and residents' health. In the figure, the horizontal axis is the air quality index, the vertical axis is the average number of medical visits of residents, and the scatter in each year is connected with the centroid of the year. The first notation for the year is the point of the average number of visits of residents and the AQI in different years. The second notation for the year is the line connecting the points of the same year to the centroid of the same year.

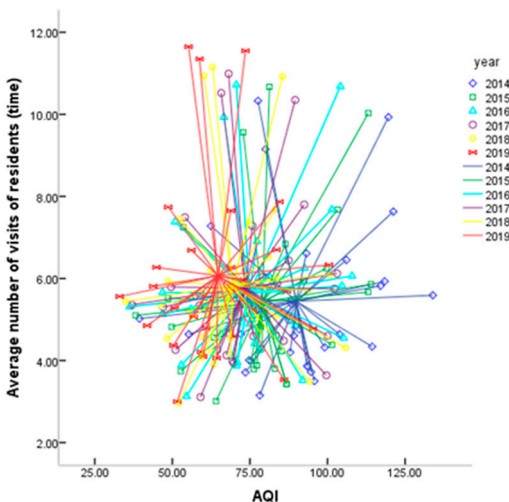

**Figure 5.** The relationship between air quality and residents' health.

Figure 5 shows that the centroid point changes from the lower right to the upper left over time from 2014 to 2019. As the air quality index decreases, the average number of residents' visits to the doctor increases. This is mainly because, in 2013, large-scale continuous haze occurred in China, which had a great impact on the travel and health of residents. Even though the air quality has been greatly improved, if residents feel slightly unwell, most will go to the hospital to determine the reason. This shows that residents are paying increasing attention to their health. It also shows that China's medical security level is improving.

### 4.3.3. Analysis of Resident Health Equation Result

Formulas (3) and (4) were systematically estimated, and Table 1 shows the estimation results. From Model 1, for every 1 unit increase in the air quality index, the average number of visits of residents increases by 0.0426 units. The impact of air quality on residents' health is significant. This shows that the health level of residents decreases as the AQI rises, which is consistent with the conclusion of Lu [63]. Each unit increase in the AQI of a surrounding province decreases the average number of medical visits of residents by 0.0083 units. The impact of the air quality index of surrounding provinces on residents' health is significant. The increase in the surrounding air quality index will cause residents to take note of the surrounding air quality. Although the local air quality is poor compared with the surrounding air quality, it is considered that the local relative environmental quality is better, which has a "soothing effect" by lessening people's concerns. The average number of visits of residents in the surrounding provinces increases by 1 unit, which will increase the average number of local residents by 0.1115 units, and the impact is significant. The increase in air pollution in surrounding provinces shows an increase in the average

number of visits of residents in the province. Due to the significant "demonstration effect" in the neighborhood, the health of residents in this province will be affected.

**Table 1.** Results of spatial simultaneous equations.

| Peh | Resident Health Equation | AQI | Air Quality Equation |
|---|---|---|---|
| | Model 1 | | Model 2 |
| AQI | 0.0426 *** | Peh_01 | −1.0922 * |
| | (5.69) | | (−1.95) |
| wAQI | −0.0083 *** | wAQI | 0.1117 *** |
| | (−5.58) | | (7.10) |
| wPeh | 0.1115 *** | Eci | 10.4944 *** |
| | (5.43) | | (3.08) |
| Pup | 0.0943 *** | Pgdp | −0.0002 * |
| | (9.81) | | (−1.96) |
| Dgv | 0.0006 *** | Psi | 0.5188 *** |
| | (3.62) | | (3.10) |
| Erg | $2.26 \times 10^{-7}$ | Tei | −0.0213 ** |
| | (1.09) | | (−2.50) |
| cons | −3.6376 *** | cons | −2.5853 |
| | (−5.28) | | (−0.28) |
| obs | 180 | obs | 180 |
| AIC | 612.260 | AIC | 1521.043 |
| BIC | 634.611 | BIC | 1549.780 |

Note: *, **, and *** indicate significance at the 10%, 5%, and 1% levels, and the parentheses are the t statistics.

An increase in the proportion of the urban population by 1 unit will increase the average number of visits of residents to health facilities by 0.0943 units, and the proportion of the urban population has a significant impact on residents' health. This shows that urban residents pay more attention to physical health than rural residents. Every increase of 1 unit of domestic waste removal and transportation will increase the average number of residents' visits by 0.0006 units, and the impact of domestic waste removal volume on residents' health is significant. The positive role of domestic waste removal and transportation has not been brought into play. Ecological restoration treatment has a non-significant effect on the average number of visits by residents. China has not reached the expected level in terms of ecological restoration and governance.

### 4.3.4. Analysis of Air Quality Equation Result

From Model 2, the average number of residents' visits to the doctor in the current period increases by 1 unit, and the air quality index in the next period decreases by 1.0922 units. When the health level of residents declines, government departments will strengthen environmental regulations and control air pollution, and the air quality will gradually improve in the next period. When the air quality index of the surrounding provinces increases by 1 unit, the local air quality index will rise by 0.1117 units. The air is fluid. When the AQI of the surrounding provinces is high, the local AQI will be higher due to the influence of the surrounding provinces. The air quality of different provinces is spatially correlated, which is consistent with the spatial correlation network of the air quality index above.

An increase of 1 unit in energy consumption intensity increases the AQI by 10.4944 units. Energy consumption remains a major culprit of air quality. An increase of 1 unit of per capita GDP will reduce the AQI by 0.0002 units. With an improvement in the quality of

life, people are more capable of improving air quality. An increase in the proportion of secondary industry by 1 unit will increase the AQI by 0.5188 units. Polluting enterprises in the secondary industry are still the target of national governance. Investment in science and technology by 1 unit will reduce the AQI by 0.0213 units. China's scientific and technological innovation has transformed from red innovation with the goal of making profits to green innovation with the goal of managing the environment. Technological innovation has begun to play an active role. All the above control variables have a significant effect on residents' health.

### 4.3.5. Robustness Test

Three methods can be used to verify whether the results in Table 1 are robust. The first is to replace variables, the second is to increase or decrease samples, and the third is to change the measurement method. This paper used the first method, and residents' health was measured by the case fatality rate (%). According to Formulas (3) and (4) for systematic estimation, Table 2 shows the estimation results. From Tables 1 and 2, the estimated results of the core explanatory variables have the same sign, but there are slight differences in the magnitude of the value. The symbols of domestic waste removal and per capita GDP changed slightly. Overall, the estimation results are robust.

**Table 2.** Impact of air quality on case fatality rate.

| Peh | Resident Health Equation | AQI | Air Quality Equation |
| --- | --- | --- | --- |
|  | Model 3 |  | Model 4 |
| AQI | 0.0032 *** | Peh_01 | −17.1567 ** |
|  | (2.58) |  | (−2.52) |
| wAQI | −0.0008 *** | wAQI | 0.0903 *** |
|  | (−4.41) |  | (7.26) |
| wPeh | 0.0970 *** | Eci | 16.0108 *** |
|  | (2.99) |  | (4.95) |
| Pup | 0.0156 *** | Pgdp | 0.0002 *** |
|  | (8.55) |  | (3.19) |
| Dgv | −0.0001 | Psi | 0.4520 ** |
|  | (−1.62) |  | (2.54) |
| Erg | $7.68 \times 10^{-8}$ * | Tei | −0.0133 |
|  | (1.87) |  | (−1.58) |
| cons | −0.6151 *** | cons | 7.9406 |
|  | (−5.03) |  | (0.93) |
| obs | 180 | obs | 180 |
| AIC | −15.229 | AIC | 1500.406 |
| BIC | 7.112 | BIC | 1529.142 |

Note: *, **, and *** indicate significance at the 10%, 5%, and 1% levels, and the parentheses are the t statistics.

## 5. Conclusions and Suggestions

### 5.1. Conclusions

First, China's air quality presents a typical spatial correlation network structure. The network structure has changed from complex to simple, and China's air quality has improved significantly. From 2014 to 2019, air pollution appears to have shifted from the east to the center of the country. In 2019, the average number of spatial correlation subjects of air quality in different provinces was more than four. Second, an increase in the local AQI will increase the average number of residents' visits to doctors. An increase in the

surrounding air quality index will cause a decline in the average number of visits to a doctor by residents. This shows that both absolute air quality and relative air quality have an impact on resident health. Third, a decline in the health of residents in the previous period will prompt the government to adopt various pollution reduction policies, which will improve air quality in the current period. The intensity of environmental regulation helps to reduce air pollution and protect residents' health [64]. This shows that various environmental regulation policies implemented by the government can reduce air pollution and improve the biological environment.

*5.2. Suggestions*

The first suggestion is to ensure the equality of health care. Air pollution has strong mobility and externalities. The adverse effects of air pollution on health do not fully comply with the principle of "who pollutes, who controls, who bears" and is manifested in the widespread diffusion of pollution. There is regional heterogeneity in China's level of economic development. The main contradiction in Chinese society at this stage will lead to underdeveloped areas and poor groups bearing higher costs of air pollution and health damage. Ensuring equal access to health care can achieve social welfare equity.

The second suggestion is to implement the integration of air treatment. The whole country should implement uniform air pollution control standards, so that when companies in economically developed areas do not meet local air governance guidelines, they cannot relocate even if they want to move to economically underdeveloped areas. With the same guidelines in all places, companies could only move to underdeveloped areas after implementing green upgrades. Green industry upgrades and transfer costs would make companies consider reducing air pollution as one of their survival goals, no matter whether they were in the local area or in the transfer area. The implementation of air treatment integration would improve absolute air quality and relative air quality to achieve the purpose of air pollution control.

The third suggestion is an environmental protection policy that keeps pace with the times. The previous economic development model produced air pollution that was hazardous to residents' health, and the damage to human capital further constrained sustainable economic development. The old path of "pollution first, treatment later" is no longer suitable for China's current situation. The environmental protection system and policy system must keep pace with the times; it must be constantly revised and improved based on environmental changes and people's livelihood issues, and it must be effective. According to General Secretary Xi, the strictest system, the strictest rule of law, and improving the intensity of environmental regulation can provide a reliable guarantee for the construction of an ecological civilization.

**Author Contributions:** H.H. and Z.L. are the co-first authors. Z.L. is the corresponding author. H.H. is responsible for the methodology and software. Z.L. is mainly responsible for writing and revising. All authors have read and agreed to the published version of the manuscript.

**Funding:** This research was supported by the National Social Science Fund of China (No. 18BJY081).

**Institutional Review Board Statement:** This study did not cause harm to humans or animals, and only studied the effects of air quality on the residents' health.

**Informed Consent Statement:** Not applicable.

**Data Availability Statement:** The raw data used to support the findings of this study. The raw data comes from National Bureau of Statistics of China. http://www.stats.gov.cn/.

**Acknowledgments:** Thanks to the editors and reviewers for their comments; our article has been greatly improved. Moreover, we would like to thank Hebei GEO University (KJCXTD-2022-02) for providing us with the necessary financial support.

**Conflicts of Interest:** The authors declare no conflict of interest.

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
