# Peer review of "Air Quality and Residents’ Health in China: An Empirical Analysis Based on Spatial Simultaneous Equations"

_applsci, doi:10.3390/app122311897_

Round 1
Reviewer 1 Report
I have some suggestions:
Fig1-4 : for better visibility, a legend for AQI should be entered.
Fig 5. Explain in detail why are there two different notations for the years? In the figure, does AQI represent the average on the country? I suggest to analyze the correlation in the most important cities in China.
Author Response
Fig1-4 : for better visibility, a legend for AQI should be entered.
Response:
Thanks deeply for your suggestion. We added a legend(AQI) for Figs1-4. The relevant content is shown in article.
Fig 5. Explain in detail why are there two different notations for the years? In the figure, does AQI represent the average on the country? I suggest to analyze the correlation in the most important
Response:
Thanks for your comment. The first notation for the year is the point of the average number of visits of residents and AQI in different years. The second notation for the year is the line connecting the points of the same year to the centroid of the same year.In the figure, AQI represents the average air quality of each province. We focus our analysis on the change of the centroid. The relevant content is shown as follows.
Figure 5 shows that the centroid point changes from the lower right to the upper left over time from 2014 to 2019. As the air quality index decreases, the average number of residents' visits to the doctor increases. This is mainly because, in 2013, large-scale continuous haze occurred in China, which had a great impact on the travel and health of residents. Even though the air quality has been greatly improved, if residents feel slightly unwell, most will go to the hospital to determine the reason. This shows that residents pay increasing attention to their health. It also shows that China's medical security level is improving.
Reviewer 2 Report
It is interestiong paper. However, some edinting efforts should be done before publishing. Some suggestions are given below.
Line 88. Pls, clarify The second
Lines 105-107. Pls, clarify meaning of the statement
Lines 474-477 The sentence is unfinished (and must be.?)
Author Response
It is interestiong paper. However, some edinting efforts should be done before publishing. Some suggestions are given below.
Line 88. Pls, clarify The second
Response:
Thanks for your helpful feedback. We have revised The second.
Lines 105-107. Pls, clarify meaning of the statement
Response:
Thanks for your insightful comment. We clarified the meaning of the statement. The relevant content is shown as follows.
The health consumption expenditure of rural residents is less than urban residents, the health consumption expenditure of male residents is less than female residents, and the health consumption expenditure of less-educated residents is less than highly-educated residents (Zhao et al.,2021).
Lines 474-477 The sentence is unfinished (and must be.?)
Response:
Thanks for your helpful feedback. We finished the sentence. The relevant content is shown as follows.
The environmental protection system and policy system must keep pace with the times; it must be constantly revised and improved based on environmental changes and people's livelihood issues and must be effective.